# Impact of Manufacturing Agglomeration on the Green Innovation Efficiency—Spatial Effect Based on China’s Provincial Panel Data

**DOI:** 10.3390/ijerph20054238

**Published:** 2023-02-27

**Authors:** Shan Xu, Yu Zhang

**Affiliations:** School of Economics, Hangzhou Dianzi University, Hangzhou 310018, China

**Keywords:** green innovation efficiency, industry heterogeneity, manufacturing agglomeration, regional heterogeneity, spatial Durbin model

## Abstract

Improving the efficiency of green innovation has become an urgent issue in the transformation of manufacturing industries in most developing countries within the context of increasing resource scarcity and environmental constraints. As an important feature of manufacturing development, agglomeration also plays a substantial role the promotion of technological progress and green transformation. Taking China as an example, this paper investigates the spatial impact of manufacturing agglomeration (MAGG) on green innovation efficiency (GIE). We first measure the level of MAGG and GIE in 30 provinces (autonomous regions and municipalities) in China during the period from 2010 to 2019, and then we utilize the spatial Durbin model in order to empirically test the spatial effect and heterogeneity based on theoretical analysis. The findings demonstrate that (1) the overall GIE in China has maintained a steady increase, and the level of MAGG slowly decreased from 2010 to 2019 with characteristics of obvious regional non-equilibrium and spatial correlations; (2) MAGG has a significant effect on the improvement of GIE nationally; (3) under the constraints of regional heterogeneity, the impacts of MAGG on GIE show significant differences between eastern, central and western China; (4) in terms of industry heterogeneity, high-tech MAGG can significantly enhance local GIE, while the indirect effect of non-high-tech MAGG is significantly negative. Our findings not only contribute to the advancement of studies pertaining to industry agglomeration and innovation, but also present policy implications for China and the world at large in terms of the development of high-quality and green economy.

## 1. Introduction

Rapid economic development and simultaneous environmental deterioration represents a common paradox faced by China and most developing countries. China’s economy has grown by leaps and bounds since 1978, and it reached a gross domestic product (GDP) of US$18 trillion in 2022. However, with the declining marginal returns of traditional factors and the optimization of industrial structures, this rapid mode of development, which relies heavily upon high energy consumption, has brought severe practical problems to China. Primary energy consumption in China reached 4.98 billion tons of standard coal and carbon dioxide emissions reached 9.9 billion tons in 2020, accounting for 26% of global energy consumption and 31% of global carbon dioxide emissions, respectively. As the second-largest economy in the world, China is trying to break away from the crude economic growth pattern of the past and gradually move towards a stage of high-quality economic development. Within the context of reaching the limits in terms of resource and environmental bearing capacity for economic growth, it is imperative to seek a path of green transformation, thereby achieving synergistic development of the economy and the environment [1]. As a conceptual combination of green practices and innovation, green innovation plays a pivotal role in solving environmental problems, stimulating new dynamics of economic growth, and expediting the extension of industrial chains [2]. Hence, the key to achieving green economic transformation rests in the improvement of green innovation efficiency (GIE) [3].

China is the largest developing country; however, its manufacturing industry is still operating in the mode of high input, high consumption, and high pollution accompanied with low energy efficiency. Taking 2019 for example, the manufacturing industry contributed 27% of China’s GDP, but it consumed about 57% of the country’s total energy consumption and it generated about 40% of the total carbon emissions. Meanwhile, it was clearly stated in the “Fourteenth Five-Year Industrial Green Development Plan” that China should accelerate the green transformation and upgrading of the manufacturing industry. Thus, it is urgent to quicken the development of green innovations in the manufacturing industry, which is undoubtedly the most important pillar of industry in China. In view of representativeness and typicality, it is believed that China is an appropriate sample for research.

Historical experience shows that manufacturing industrial agglomeration (MAGG) has become one of the most dynamic spatial organization methods, and that it has played a major role in local economic development during the continuous improvement of modern market economies [4]. Theoretically, industrial agglomeration is conducive to the flow of production factors between regions and the division of labor along the industrial chain. MAGG contributes to the separation and spatial agglomeration of inputs, production, and to research and development (R&D), which possibly results in the reduction of inputs and the improvement of resource allocation efficiency [5]. In addition, reasonable agglomeration can achieve large-scale, intensive, and efficient production, thus enhancing green economy development [6]. 

Although the factors affecting GIE—such as environmental regulation, opening-up, and government support—have been examined in massive studies [7,8,9], rarely has attention been paid to the perspective of MAGG. Compared with the existing research, this paper mainly contributes in four aspects: (1) it proposes an impact mechanism between MAGG and GIE; (2) it measures the level of MAGG and GIE in China and analyzes the temporal and spatial trajectories of evolution; (3) it fully considers the spatial characteristic of variables, adopts the spatial panel Durbin model to test the impacts of MAGG on GIE in China, and uses instrumental variables to overcome the possible regression bias; and (4) it examines the heterogeneity of the impact from both regional and industrial perspectives. The remainder of the paper is organized as follows: Section 2 presents a literature review and theoretical analysis; Section 3 measures the levels and describes the evolutionary characteristics of both MAGG and GIE; Section 4 introduces empirical models and the selection of variables; Section 5 demonstrates and explains all of the empirical results; and Section 6 discusses conclusions and resultant policy implications.

## 2. Literature Review and Theoretical Analysis

### 2.1. Literature Review

Green innovation is also known as eco-innovation, environmental innovation, etc. [10]. With attributes of both green policies and innovation, it is an effective means to break the constraints of resources and the environment and simultaneously promote sustainable development [11]. According to the existing literature, green innovation includes the reduction of environmental impact, the introduction of environmental performance and environmental innovation, and the improvement of environmental performance [12]. Referring to the viewpoint of Chen (1999) [13], green innovation is regarded, in this paper, as innovation which balances economic benefits and environmental effects. Industrial agglomeration refers to the spatial clustering of numerous related enterprises and their supporting organizations in a particular area, leading to the formation of a strong competitive advantage [14].

Many scholars have conducted studies on the economic and technological effects of industrial agglomeration. Yuan et al., (2022) concluded that MAGG affected green development in three ways—upgrading labor, upgrading the industrial structure, and technological innovation effect—and they found an inverted U-shaped relationship between MAGG and green development in the Yangtze River Economic Zone of China [15]. Using panel data from the Yangtze River Delta urban agglomeration in China as a sample, Sun et al., (2022) argued that an increase in industrial specialization agglomeration would inhibit the improvement of urban green economic efficiency, while a “U”-shaped relationship was found between industrial diversification agglomeration and urban green economic efficiency [16]. Zhang and Liu (2023) also found that urban industrial agglomeration could promote the economic growth of the agglomeration area, and that the technological innovation brought about by said agglomeration could effectively promote economic development in the short term, though the sustainability is not high [17]. Xu and Zhou (2022) concluded that industrial agglomeration would improve environmental efficiency through scale, technological, and social effects [18]. Zhao (2020) found that the interactions between enterprises and the surrounding environment in the process of industrial agglomeration would further affect the efficiency of regional green innovation [4].

Currently, there are three main views on the relationship between industrial agglomeration and innovation efficiency. First, industrial agglomeration can improve innovation efficiency. Peng et al., (2011) discovered that industrial agglomeration had a substantial driving effect on innovation efficiency by exploring intra-industry knowledge spillover (MAR externality) and inter-industry knowledge spillover (Jacob’s externality) [19]. Using data from the Spanish footwear industry, Ruiz-Ortega (2016) found that firms in industrial agglomeration had greater advantages in terms of profitability and innovation when compared to those not engaged in industrial agglomeration [20]. Zeng and Wang et al., (2019) adopted a spatial econometric model and found that innovation efficiency and the agglomeration of productive services contributed to each other with significant spatial spillover effects [21]. Second, industrial agglomeration can inhibit innovation efficiency. Based on provincial panel data on the electronics and communication equipment manufacturing industry, Lai and Wang et al., (2022) utilized panel stochastic frontier regression and found that industrial agglomeration hinders the improvement of regional innovation efficiency [22]. From an industry perspective, Zhang et al., (2016) applied the data from 29 provinces and cities in China and discovered that the relationship between industrial agglomeration and industrial innovation efficiency was greatly negative [23]. With regards to the relationship between MAGG and GIE, Ren et al., (2020) found that MAGG always produced an inhibitory effect on the improvement of GIE in both the long and short term [5]. Also, Xie et al., (2019) used the Yangtze River Delta city cluster as a research sample and discovered that MAGG was not conducive to the improvement of innovation efficiency after considering the knowledge spillover of industrial agglomeration and the externalities of imitative innovation [24]. Third, the relationship between industrial agglomeration and innovation efficiency is non-linear, and the impact depends upon the combined result of both the positive and negative externalities introduced by industrial agglomeration. As discovered by Yuan and Guo et al., (2018), the relationship between MAGG and technological innovation was inverted “U”-shaped, with the eastern region on the right side of the curve, and the central and western regions of the country on the left side of the curve [25]. Using the panel Tobit model and threshold model, Li and Zeng (2021) found that the clustering of high-tech industries and GIE both show a decreasing trend from the east to the west in terms of spatial pattern [26]. Additionally, Zhu et al., (2021) discovered that the impact of industrial agglomeration on GIE varied under the different intensities of environmental regulation and foreign-invested industries [27].

In summation, scholars have carried out a large number of studies on industrial agglomeration and innovation efficiency. However, there are still some shortcomings: first, existing studies seldom directly focus on the systematic analysis of MAGG and GIE; second, most relevant research previously conducted ignores the spatial spilling effect and endogenous issues of variables, resulting in biased regression results; third, insufficient attention has been paid to the heterogeneity between MAGG and GIE from both the regional and industrial perspectives. Therefore, there is an opportunity for our study to make some improvements.

### 2.2. Theoretical Analysis

According to the existing literature and historical experience, it is regarded that MAGG has both positive and negative impacts on GIE (Figure 1).

#### 2.2.1. Positive Externalities

(1)Spillover Effect. Since there is time-lag in the transmission of knowledge and technology, and given that the spillover effect usually decreases as the spatial distance increases, distance is thus an important obstacle to the spillover of knowledge and technology [28]. As tacit knowledge can only be acquired by face-to-face communication, firms prefer to form ‘industrial organizations’ in order to share knowledge and technology. When there is a breakthrough in energy-saving and emission-reducing products or technologies in the cluster, enterprises within the cluster can access such knowledge in a timely and cost-effective manner, thus reducing the cost of self-exploration and resource utilization and improving the overall GIE.(2)Resource-Sharing Effect. By clustering manufacturing enterprises into agglomerations, better integration and more flexible mobility of production factors such as capital, land, and labor can be ensured. Moreover, by sharing infrastructure development and labor markets, enterprises can avoid redundant construction, thereby saving more on costs for innovation and increasing regional GIE. As the overall quality of the regional economy can be improved by agglomeration, the government and relevant financial institutions tend to provide the established manufacturing parks with policy incentives and supports such as infrastructure construction so as to optimize the external conditions for enterprise development. Simultaneously, agglomeration can also serve as a platform for face-to-face exchange, thereby attracting different institutions, enterprises, and talents with green and innovative ideas to cooperate, accelerating the spillover and dissemination of green technologies, increasing the flow of production factors, and, ultimately, promoting efficient green innovation.(3)Other Effects. Specialized division and economies of scale can also help to increase the GIE. The agglomeration of a large number of enterprises forms a complete industrial chain, and the specialized division formed by trusting enterprises in the agglomeration area will reduce the intermediate inputs, greatly improve the utilization of resources, decrease the learning cost, enhance the professionalism and proficiency of the workforce, and strengthen the knowledge and technology spillover effect of during exchange and cooperation. This can promote technological progress, accelerate product innovation, and ultimately expedite product innovation. Meanwhile, by sharing infrastructure, specialized services, science and technology, and talent markets within the agglomeration area, the average cost to enterprises can be reduced, and economies of scale can be created. Furthermore, specific manufacturing clusters are also beneficial to the implementation of large-scale pollution control, thereby sharing pollution control costs, facilitating regulation, and realizing green manufacturing development.

#### 2.2.2. Negative Externalities

(1)Crowding Effect. This refers to the possibility of numerous manufacturing industries gathering together for production inside a limited area with the scale exceeding the carrying capacity [29]. The crowding effect can cause problems such as factor scarcity and environmental pollution. On the one hand, the expansion of MAGG scale will inevitably lead to a surge in demand for production resources such as land and capital, the oversupply of resources will cause the price of resources to rise, and this scarcity of resources will directly generate an increase in the production costs of enterprises, thus reducing the R&D costs of green innovation, which is not conducive to the improvement of GIE. On the other hand, the agglomeration of manufacturing industries will bring about a massive concentration of population, and the resultant increase in people’s demands for food, clothing, housing, and transportation will in turn cause a large amount of domestic pollutants to be emitted and not easily managed, thus further deteriorating the environment, which is also not beneficial to the promotion of GIE [30].(2)Excessive Competition Effect. Due to limited geographical space, increased agglomeration may also cause excessive competition among enterprises, manifested by vicious price competition and imitation innovation. Considering the inherent market share, enterprises will successively lower their prices in order to remain competitive, thus resulting in the inability of enterprises to make reasonable profits and normal returns. This process may also compel enterprises to reduce R&D expenditure in order to maintain operations, which would result in a decrease in the GIE of the whole park. Furthermore, in areas with weak intellectual property protection, clustered technological spillover and personnel mobility will give rise to imitation. Due to the short-term benefits brought about by imitation, the recipients of knowledge spillover may show no incentive to innovate. If the innovation investment even cannot be recovered, innovational incentive will be heavily reduced, which hampers the enhancement of GIE as well [31,32].

## 3. Measurement and Analysis of MAGG and GIE

### 3.1. Measurement of GIE

GIE focuses on the greening degree of innovation efficiency, and it is an evaluation of innovation quality after considering resource utilization and environmental pollution. Green innovation is an important symbol of sustainable development for a region during times of factor scarcity and environmental degradation, and innovation efficiency is an important indicator of the effect of input–output allocation within green innovation, with the efficiency value reflecting the effects of the resource allocation behavior of each subject [33]. Therefore, indicators are constructed in this paper in order to measure the GIE from the input–output perspective, and the Super-SBM model proposed by Tone (2002) [34] is adopted with the purpose of effectively dealing with non-expected outputs and comparing different effective decision-making units.

Supposing that there are n decision units with m  innovative inputs each, and $x=\left(x_{1},x_{2},\ldots ,x_{n}\right)\epsilon R^{m\times n}$, then, $k_{1}$ desired outputs $y^{g}=\left(y_{1}^{g},y_{2}^{g},\ldots ,y_{n}^{g}\right)\epsilon R^{k1\times n}$ and $k_{2}$ non-desired outputs $y^{b}=\left(y_{1}^{b},y_{2}^{b},\ldots ,y_{n}^{b}\right)\epsilon R^{k2\times n}$ are produced, where $X>0,\;\;Y^{g}>0,\;\;Y^{b}>0$. Hence, the set of production possibilities containing non-expected outputs is constructed as $P=\{\left(x,y^{g},y^{b}\right)|x\geq X\lambda ,y^{g}\leq Y^{g}\lambda ,y^{b}\geq Y^{b}\lambda ,\lambda \geq 0\}$, with Super-SBM model established as below.
(1)$$\rho^{*}=\min\frac{\frac{1}{m}\sum_{i=1}^{m}\frac{\bar{x_{i}}}{x_{i0}}\;}{\frac{1}{k_{1}+k_{2}}\left(\sum_{r=1}^{k1}\frac{{\bar{y}}_{r}^{g}}{y_{r0}^{g}}+\sum_{r=1}^{k1}\frac{{\bar{y}}_{1}^{b}}{y_{r0}^{b}}\right)}$$

The constraints are $\bar{x}\geq \sum_{i=1,\neq 0}^{n}\lambda_{i}x_{i}$, ${\bar{y}}^{g}\leq \sum_{i=1,\neq 0}^{n}\lambda_{i}y_{i}^{g}$, ${\bar{y}}^{b}\geq \sum_{i=1,\neq 0}^{n}\lambda_{i}y_{i}^{b}$, $\bar{x}\geq x_{0},{\bar{y}}^{g}\leq y_{0}^{g},{\bar{y}}^{b}\geq {\bar{y}}_{0}^{b},\lambda >0$, where λ is the weight vector; $\bar{x},{\bar{y}}^{g},{\bar{y}}^{b}$ refer to the slack vectors of inputs, desired outputs, and non-desired outputs; and $\rho^{*}$ represents the target efficiency. The larger the value of $\rho^{*}$, the more efficient the decision unit is.

All of the evaluation indicators determining GIE are exhibited in Table 1. The full-time equivalent of R&D personnel and R&D input, new product development expenditure, and the energy consumption of each province (autonomous regions and municipalities) are selected as input indicators. The number of green patents granted and the new product sales revenue of each province (autonomous regions and municipalities) were chosen as the desired output indicators, and CO_2_ emissions were deemed to be non-desired output indicators. 

A total of 30 provinces (autonomous regions and municipalities) in China were selected as the main research subjects (excluding Tibet, Hong Kong, Macau, and Taiwan). In order to avoid the uncertain influences of COVID-19 and in view of the availability as well as consistency of data, the selected time span was from 2010 to 2019. For the individual missing data, interpolation methods were used to fill in the gaps. The samples were further divided into the eastern region, the central region, and the western region for analyses (the measurement regions are divided in accordance with the announcement from the National Bureau of Standards (NBS). The eastern region includes Beijing, Tianjin, Hebei, Liaoning, Shanghai, Jiangsu, Zhejiang, Fujian, Shandong, Guangdong, and Hainan. The central region includes Shanxi, Jilin, Heilongjiang, Anhui, Jiangxi, Henan, Hubei, and Hunan. The western region includes Inner Mongolia, Guangxi, Chongqing, Sichuan, Guizhou, Yunnan, Tibet, Shaanxi, Gansu, Qinghai, Ningxia, and Xinjiang). As indicated in Figure 2, the GIE shows a fluctuating upward trend in China and the three regions therein during the period from 2010 to 2019. Thereof, the GIE of eastern region is consistently higher than the level of national average, and the central region is lower than the national average before 2015, though it catches up with the national average after 2015. The GIE of the western region barely fluctuates over the decade.

According to the GIE results, Beijing, Zhejiang, and Tianjin are the top three provinces (autonomous regions and municipalities) in terms of average level, while Shanxi, Inner Mongolia, and Qinghai represent the bottom three. Qinghai, Xinjiang, and Jiangxi are at the top in terms of GIE growth rate over the decade, while Hainan, Tianjin, Shanghai, Jilin, Chongqing, and Yunnan have negative growth. Figure 3 provides the radar chart of the level of GIE in these provinces (autonomous regions and municipalities), showing a spatial comparison between the year 2010 and the year 2019. As indicated, the GIE changes obviously over the course of the decade and shows an unevenness between regions.

### 3.2. Measurement of MAGG Level

The level of industrial agglomeration can be measured using various methods, such as the Herfindahl index (HHI), locational entropy, spatial Gini index, and the Elilsion and Glaeser (EG) index. Since location entropy reflects the specialization degree of an industrial agglomeration well and eliminates the factors of regional scale difference, the spatial distribution of geographical factors is able to be truly reflected. Therefore, the method of locational entropy was adopted in this paper in order to measure the degree of MAGG in China [35]. The locational entropy of industry  j in region  i is calculated with the formula below.
(2)$$magg_{ij}=\frac{x_{ij}/\sum_{i}x_{ij}}{\sum_{j}x_{ij}/\sum_{i}\sum_{j}x_{ij}}$$
where $x_{ij}$ denotes the number of people employed in industry j in region  i.

As observed from Figure 4, all of the MAGG levels showed a slow decreasing trend during 2010–2019. The ranking of the three regions was eastern region, central region, and western region, and this remained basically unchanged throughout the decade. Specifically, excepting for the eastern region where the MAGG level was always higher than the national average, the remaining two regions consistently maintained lower MAGG levels. Also, compared to the fluctuations in MAGG levels in the eastern region, those in the central and western regions were more stable. 

The top three provinces (autonomous regions and municipalities) in terms of MAGG level were Zhejiang, Jiangsu, and Shanghai, while the bottom three were Guizhou, Gansu, and Hainan. Anhui, Guangdong, and Chongqing showed significant changes in MAGG level over the 10-year period. There were also several provinces (autonomous regions and municipalities) experiencing negative growth. Figure 5 displays the changes in these provinces (autonomous regions and municipalities). As can be observed, there is a large gap between the top three and the bottom three regarding the MAGG level, and the evolutionary trend of MAGG is showing obviously regional difference.

### 3.3. Correlation Analysis between MAGG and GIE

Figure 6 shows the scatter plot of MAGG and GIE values. As the MAGG increases, the fitted line slopes to the upper right, which shows that GIE and MAGG have a significant correlation in each region over the course of the decade.

## 4. Research Design

### 4.1. Model Construction

Based on the theoretical analysis, the baseline regression model was constructed as follows:(3)$${\mathrm{GIE}}_{\mathrm{it}}=\gamma {\mathrm{MAGG}}_{\mathrm{it}}+{X_{it}}^{\prime }\phi +\alpha_{i}+\eta_{t}+\varepsilon_{it}$$
where i  represents the province (autonomous regions and municipalities), t represents the year, ${\mathrm{GIE}}_{\mathrm{it}}$ represents GIE, ${\mathrm{MAGG}}_{\mathrm{it}}$ represents MAGG level, γ is an explanatory variable coefficient, ${X_{it}}^{\prime }$ is a control variable, ϕ is the coefficient vector of a control variable, $\alpha_{i}\;\mathrm{and}\;\eta_{t}$ are fixed by region and year, respectively, and $\varepsilon_{it}$ is a random disturbance term.

Due to the possible spatial correlation between MAGG and GIE, a spatial econometric model was further adopted in order to conduct empirical research. The equation for the spatial econometric model is as follows:(4)$${\mathrm{GIE}}_{\mathrm{it}}=\rho W_{ij}{\mathrm{GIE}}_{\mathrm{it}}+\beta_{1}{\mathrm{MAGG}}_{\mathrm{it}}+\beta_{2}X_{it}+\theta_{1}W_{ij}{\mathrm{MAGG}}_{\mathrm{it}}+\theta_{2}W_{ij}X_{it}+\alpha_{i}+\eta_{t}+\varepsilon_{it}.\;$$
(5)$$\varepsilon_{it}=\lambda W_{\varepsilon_{it}}+\mu_{it}.\;$$
where ρ is the coefficient of the spatial lag term of GIE, $\beta_{i}$ refers to all of the parameters to be estimated, $X_{it}$ is the control variable, $\theta_{i}$. is the spatial spillover coefficient of the independent variable coefficient, $\varepsilon_{it}$ is the random disturbance term, and $\mu_{it}$ is the random error term subject to normal distribution. The spatial measurement models used were the spatial error model (SEM), spatial autoregressive model (SAR), and spatial Durbin model (SDM). If $\lambda =\theta_{1}=\theta_{2}=0$, then the econometric model is the SAR model. If $\rho =\theta_{1}=\theta_{2}=0$, then it is the SEM model. If λ=0, then it is the SDM model.

$W_{ij}$ refers to the spatial weight matrix. Considering the possible autocorrelation of GIE among provinces (autonomous regions and municipalities), the GIE of one region may be influenced by neighboring regions. Also, considering the different geographical distances between regions, the degree of impact may also be different. Hence, the inverse of the spatial geographical distance is applied in order to construct the spatial geographical distance matrix $W^{d}$, where $d_{ij}$ denotes the distance between two regions calculated based on latitude and longitude data, as follows:(6)$$W^{d}=\left\{\begin{array}{c} \frac{1}{\left|d_{ij}\right|},\;i\neq j\\ 0,\;i=j \end{array}\right.$$

### 4.2. Description and Descriptive Statistics of Variables

Referring to the existing research, several control variables were introduced in this paper, namely environmental regulation (ENV), industrial structure (IS), opening-up (FD), economic development (GDP), and transport infrastructure (TRA). These variables are exhibited in Table 2, and descriptive statistics for the main variables are exhibited in Table 3.

## 5. Empirical Analysis

### 5.1. Baseline Regression Result

Table 4 shows the baseline regression results of the impact of MAGG on GIE. With the gradual introduction of control variables, MAGG always has a significant positive impact on GIE, that is, reasonable MAGG contributes to the improvement of GIE.

### 5.2. Endogeneity and Instrumental Variables

However, it is usually believed that an endogenous relationship exists between MAGG and GIE. MAGG affects GIE, but provinces with high GIE also show high levels of agglomeration density—that is, there may be a reverse causality between MAGG and GIE. Although the factors affecting MAGG have been considered as comprehensively as possible during the model construction process, the missing variables still cannot be effectively controlled. Endogeneity problems can lead to bias in the estimation of results.

This paper uses the instrumental variable (IV) and two-stage least-square (2SLS) methods to solve the endogenous problem. An appropriate IV should meet two requirements: correlation and exogenesis; that is, it should be related to MAGG but relatively exogenous for GIE. Scholars usually select IVs from the perspective of geography or history, because some indicators of geography are formed naturally, while some variables in history are far from the present, so it can be supposed that neither of them directly affect the explained variables in a modern economic system, thus meeting the exogenous conditions. In order to overcome the endogenous problems brought by MAGG, Duan et al., (2022) chose average ground slope and surface roughness as IVs and analyzed the influence of MAGG on labor wages [36]. Yuan et al., (2019) took whether there were railways in Chinese cities in 1933 as an IV of economic agglomeration and thereby measured the impact of MAGG on green development from a historical perspective [15].

Based on the method of Lin and Tan (2019) [37], this paper takes the topographic relief (GIS) of each province (autonomous regions and municipalities) as an IV of MAGG. The higher the GIS, the lower the MAGG level. From the perspective of exogenesis, as a natural index, topographic relief does not directly affect GIE and is relatively exogenous to it. Meanwhile, in terms of correlation, the degree of relief is closely related to plant construction, infrastructure construction, population concentration, and economic development. Data on GIS came from the measurement results of Feng et al., (2007) [38]. Referring to Wu and Shao (2016) [39], the population size (POP) of each province (autonomous regions and municipalities) in 1990 was selected as the other IV of MAGG from the historical dimension, with data originating from the China Statistical Yearbook. The sample period of the study in this paper was from 2010 to 2019, which is more than 20 years after 1990. Such a long interval ensures that it will not be correlated with the residual term of the model. In addition, population is an important driving factor of economic growth, and the POP can reflect the labor agglomeration situation of a region at that time as well as the strength of the agglomeration externality, especially when closely related to the development of manufacturing industry.

Columns 2 and 3 of Table 5 present the re-estimation results of the model using the IVs GIS and POP. After adding all of the control variables, the F-test value at the first stage was more than 10, which conforms to empirical rule, and indicates that there is no weak IV problem. Based on the second-stage regression result, after controlling the endogenous problem, MAGG still shows a significant optimization effect on GIE.

### 5.3. Spatial Autocorrelation Test

Owing to effects such as knowledge and technology diffusion, the MAGG and GIE of one region can be inevitably influenced by those of other regions. In this section, the Global and Local $Moran^{\prime }s\;I$ indexes (measures of spatial autocorrelation) are applied in order to measure the spatial correlation of the MAGG and GIE in China. 

#### 5.3.1. Global Spatial Autocorrelation Test and Results

The Global $Moran^{\prime }s\;I$ index of MAGG and GIE is measured and calculated as below.
(7)$$Global\;Moran^{\prime }s\;I=\frac{\sum_{i=1}^{n}\sum_{j=1}^{n}W_{ij}\left(X_{i}-\bar{X}\right)\left(X_{j}-\bar{X}\right)}{\sum_{i=1}^{n}\left(X_{i}-\bar{X}\right)/n}\times \frac{1}{\sum_{i=1}^{n}\sum_{j=1}^{n}W_{ij}}$$

The value of Global Moran’s *I* index is [−1, 1], greater than 0, indicating the existence of spatial autocorrelation. The closer the value is to 1, the higher the concentration of spatial units with similar GIE or MAGG in each region. If the value is less than 0, then there shall be negative spatial correlation. The closer to −1 the value is, the greater the difference in each region, or the less the distribution. The closer the value is to 0, the more randomly distributed in each region is, and there is no spatial autocorrelation.

The results are exhibited in Table 6. As indicated, the Global Moran’s *I* index of MAGG and GIE in China has mainly hovered at 5% in most years, suggesting that MAGG and GIE are distributed with a significantly positive spatial correlation rather than randomly distributed among provinces (autonomous regions and municipalities) in China. These results further illustrate that areas in close geographical proximity are more likely to show mutual influence. Hence, it is necessary to analyze the impact of MAGG on GIE with a spatial econometric model.

#### 5.3.2. Local Spatial Autocorrelation Test and Results

The local Moran’s *I* index is used to reflect the spatial correlation of variables in local regions of China, which is calculated as follows.
(8)$$Local\;Moran^{\prime }s\;I=\frac{\left(X_{i}-\bar{X}\right)\sum_{i=1}^{n}\sum_{j=1}^{n}W_{ij}\left(X_{j}-\bar{X}\right)}{\sum_{i=1}^{n}{\left(X_{i}-\bar{X}\right)}^{2}/n}$$

In order to analyze the characteristics of the local spatial distribution of GIE in China, local Moran’s *I* scatter plots of the GIE in China over the years 2010–2019 were drawn. As indicated in Figure 7, most provinces (autonomous regions and municipalities) are located within quadrants one and three, which again proves that there is a positive spatial correlation between GIE in each region. Specifically, Beijing, Tianjin, Jiangsu, and Guangdong are always located in the first quadrant, indicating a high-high (HH) agglomeration. With the passage of time, the number of provinces (autonomous regions and municipalities) in the first quadrant increases, especially those in the Yangtze River Delta and Pearl River Delta regions in eastern China, while the provinces (autonomous regions and municipalities) are central and western China is mainly in the third quadrant of low–low (LL) agglomeration. A strong local spatial agglomeration effect of GIE is revealed.

With respect to MAGG, as presented in Figure 8, most provinces (autonomous regions and municipalities) are located in quadrants one and three, exhibiting spatial autocorrelation. The provinces (autonomous regions and municipalities) located in the first quadrant show a HH agglomeration, namely Shanghai, Zhejiang, Fujian, and Jiangsu. However, different from GIE, there are relatively fewer provinces (autonomous regions and municipalities) in the first quadrant in 2019, which conforms to the results of the MAGG level measurements. Nevertheless, China’s MAGG level still exhibits a strong local spatial agglomeration effect.

### 5.4. Selection of Spatial Econometric Model and Regression Results

First, the Lagrange multiplier (LM) statistic test and robust LM statistic test were administered to the non-spatial panel model. As can be observed from the LM test results, the statistics were all significant at the 1% level, indicating the simultaneous existence of SEM and SAR, and the SDM was considered for selection. Then, further likelihood ratio (LR) tests showed that the SDM model could not be degraded to the SEM model or SAR model, and as such the SDM model was chosen. Additionally, the Hausman test results rejected the original hypothesis at the 10% level, suggesting that fixed effects should be used. Hence, the SDM model with fixed effects was ultimately selected to carry out the research.

According to the regression results shown in Table 7, the spatial autoregressive coefficient (ρ) of GIE and the spatial lag term coefficient (Wx: MAGG) of MAGG are both significant at a 5% level, indicating that there is a notable positive spatial spillover effect between of GIE and MAGG in China; GIE and MAGG in each region are influenced not only by relevant local factors, but also by the factors of neighboring region. Furthermore, the effect of MAGG on local GIE was remarkably positive at the 1% level, that is, every 1% increase in MAGG would cause a 0.093% increase in GIE. As the theoretical analysis mentions, this may possibly result from the positive externalities brought about by MAGG which promote the development of green technology by virtue of knowledge and technology spillover, the production and human resources, specialized division, and economies of scale, thereby improving the overall GIE of the region.

Each control variable also plays a different role. ENV is obviously negative at the 1% level, possibly because ENV is an additional cost and constraint imposed upon firms by the government in order to internalize the external costs, and in response to the ENV policies implemented by the government, firms increase the relevant investments in order to reduce environmental pollution, thus creating a crowding-out effect, namely that the investment in environmental management may crowd out the investment in innovation, leading to a decrease in GIE, an idea which is consistent with the research results of Liu et al., [40]. In contrast, FD and TRA are positively correlated with GIE at the 1% level, suggesting that FD is an important factor in GIE improvement, and improved TRA also contributes to GIE. Moreover, the relationship between the GDP and GIE is significantly negative, probably due to the fact that when economic development reaches a certain level, the growth of the economy leads to the redundancy of innovation inputs and undesired outputs, resulting in a loss of GIE, which is also similar to the scenario presented by Wu (2017) [41].

### 5.5. Robustness Tests

Three methods are adopted to test the robustness of the empirical results in this paper. (1) Matrix replacement: a spatial economic distance matrix is constructed for regression by using the average value of the actual GDP per capita in the observation period as an indicator of economic development. (2) Explanatory variable replacement: the HHI is utilized in to remeasure the level of MAGG. (3) Time span adjustment: the Ministry of Industry and Information Technology of China released the Industrial Green Development Plan (2016–2020) in 2016, which brought in strict regulations on high-polluting manufacturing industries and encouraged a large number of overseas acquisitions by Chinese enterprises. Therefore, data after 2016 are omitted from the regression analyses in order to avoid the potential impact of a specific year on the GIE. As observed from the regression results in Table 8, the empirical findings are stable.

### 5.6. Analysis of Direct and Indirect Effects

The effects of MAGG on GIE can be broken down into direct effects and indirect effects by using the partial differential method [42]. The direct effects represent the impact of local MAGG on local GIE. The indirect effects indicate the impact of local MAGG on neighboring regions. The total effect represents the overall impact of MAGG on GIE. As can observed from Table 9, the direct effects of the impact of MAGG were significant at a 5% level under the geographical distance weighting matrix. A 1% increase in the region’s MAGG will cause the local GIE to increase by 0.086%. The logic behind it may be that MAGG brings about knowledge and technology spillover, accelerates the diffusion of energy-saving, emissions reduction and environmental protection technologies, and provides an exchange platform for local green innovation, thereby attracting more talent with environmental protection ideas. Simultaneously, due to the sharing effect pertaining to resources and talents and the lower production costs caused by economies of scale, investment in green research and development is increased, and thus the GIE in the region is improved. However, the indirect effect is not significant; that is, the MAGG of neighboring provinces (autonomous regions and municipalities) does not have an effect upon local GIE, and the spatial spillover effect of MAGG on GIE is not effectively displayed by geographical adjacency between provinces, a situation which is similar to Wu and Wu’s research in 2021 [43].

In terms of control variables, the direct and indirect effects of ENV and GDP are both negative at the 5% level. ENV and GDP not only inhibit local GIE, but also the GIE in neighboring areas. The direct effect of IS on GIE is significantly positive, indicating that the optimization of IS is conductive to the improvement of local GIE. Additionally, the direct and indirect effects of FD and TRA on GIE are both remarkable at a 5% level, suggesting that improved FD and TRA are beneficial to GIE promotion in both local and neighboring areas.

### 5.7. Heterogeneity Tests

#### 5.7.1. Regional Heterogeneity Test

Since the level of MAGG varies from east to west, and given that there are great differences in terms of regional resource endowments, technological development, level of opening-up, and other conditions [44,45,46], samples were are further grouped by regions for testing for regional heterogeneity. The results are shown in Table 10.

The direct effect of MAGG on GIE in eastern China is significantly positive at a 1% level, with the indirect effect being notably negative at the 1% level. The most credible explanation is that the eastern region shows a remarkable lead in both the level green innovation and the internal division and cooperation in industrial agglomeration. Due to predominant economic conditions, the MAGG in the eastern region attracts many surrounding production factors in addition to absorbing local production factors in the development of green innovation technologies. As GIE is the ratio of green innovation resource supply to output in production, and the development of GIE in neighboring regions is inhibited, the indirect effect is thus shown to be negative.

The indirect effect of MAGG on GIE in central China is significantly positive, while the direct effect is insignificant, and the direct and indirect effects of MAGG in western China are both insignificant. The reasons behind this are that the central and western regions are inland with low marketisation and high transport costs, and the level of MAGG and development in these areas is much lower than that in eastern regions, resulting in the impact of MAGG has not yet begun to take effect. Nevertheless, with the deepening of the division in the industrial chain between regions, the spatial spillover effect of MAGG in the central region comes out.

#### 5.7.2. Industry Heterogeneity Test

In view of the possible heterogeneous impacts of MAGG in different manufacturing sectors on regional GIE, manufacturing industries were divided into 7 high-tech sectors and 23 non-high-tech sectors (According to the Classification of High-Tech Sectors [Manufacturing] [2017] and the Industrial Statistics Yearbook, high-tech manufacturing sectors include chemical raw materials and chemical products manufacturing, pharmaceutical manufacturing, chemical fiber manufacturing, other transport equipment manufacturing, electronic machinery and equipment manufacturing, communications equipment and computer and other electronic equipment manufacturing, instrumentation and cultural and office machinery manufacturing; non-high-tech sectors include agriculture and food processing, food manufacturing, beverage manufacturing, tobacco manufacturing, textiles, textile, clothing, and footwear manufacturing, leather, fur, feather (down) and their products, wood processing and wood, bamboo, rattan, palm and grass products, furniture manufacturing, paper and paper products, printing and reproduction of recorded media, education and sporting goods manufacturing. Additionally, there are the petroleum processing and coking and nuclear fuel processing industry, rubber and plastic products industry, non-metallic mineral products industry, ferrous metal smelting and rolling processing industry, non-ferrous metal smelting and rolling processing industry, metal products industry, general equipment manufacturing industry, special equipment manufacturing industry, automobile manufacturing industry, other manufacturing industry, waste resources and waste materials recycling industry) for further research.

Referring to Liu et al., [47], the method of locational entropy is adopted again in order to measure the agglomeration level of the high-tech and non-high-tech manufacturing sectors in all provinces (autonomous regions and municipalities). The calculation formula is as follows:(9)$$magg_{ab}=\frac{p_{ab}/p_{a}}{p_{b}/p}$$
where $p_{ab}$ represents the output value of industry *b* in region *a*, $p_{a}$ is the total output value of region *a*, $p_{b}$ is the total national output value of industry *b*, and p is the gross national product.

Figure 9 shows the trend of high-tech and non-high-tech MAGG levels during the years from 2010 to 2019. As can be seen from the figure, the level of non-high-tech MAGG is always above the level of high-tech MAGG, and the changes to both are relatively stable during these ten years. Figure 10 displays the MAGG level of representative provinces (autonomous regions and municipalities) in 2010 and 2019. In 2010, the highest and lowest levels of high-tech MAGG were for Jiangsu and Heilongjiang, respectively, and the highest and lowest levels of non-high-tech MAGG were for Shandong and Beijing, respectively. In 2019, Guangdong and Gansu were ranked first and last in terms of high-tech MAGG, while Fujian and Beijing occupied the top and bottom of non-high-tech MAGG. Generally, Guangdong, Jiangsu, Zhejiang and Shanghai were always in the lead of high-tech MAGG. Judging from the trends, Ningxia and Qinghai grew the fastest in terms of high-tech MAGG during the decade, while Guangdong decreased.

As is exhibited by the decomposition results in Table 11, the direct effect of high-tech MAGG on GIE is remarkable at the 1% level, that is, when high-tech MAGG increases by 1%, the local GIE shall rise by 0.275%. The explanation for this may be that the agglomeration of high-tech manufacturing sectors can create a large number of jobs, thereby attracting numerous talent with green innovation expertise, and that the formation of a better-quality labor market can solve the problem of supply and demand for green innovation talents. Furthermore, the agglomeration of high-tech manufacturing sectors makes it easier to disseminate knowledge and technology, causing different enterprises to learn from and imitate each other, thereby forming healthy competition and ultimately promoting the improvement of green innovation technology. The indirect effect is not notable, which may be the result of a combination of factors, such as institutions, technology, and capital. For example, high-tech manufacturing sectors, including the aerospace vehicle and equipment manufacturing industry, are related to national defense and security, with high requirements for technology and capital, resulting in a certain weakening effect on the spillover scope of high-tech MAGG and an insignificant impact on the level of GIE in neighboring places.

As to non-high-tech manufacturing sectors, the indirect effect of MAGG on GIE is notably negative, and the direct effect is not notable. The reason for this may be that most of the non-high-tech manufacturing sectors are labor-intensive and capital-intensive industries, and most of them follow a simple agglomeration pattern, the production of which primarily concentrates on product and market competition. In the case of heavy agglomeration, large amounts of production resources are consumed, leading in turn to more pollutants, all of which reduce the GIE in and around agglomerations. This is also consistent with the realities of these industries and their sectoral characteristics.

Such being the case, from the perspective of industry segmentation, there is industry heterogeneity in the impact of MAGG on GIE, and high-tech MAGG plays a more important role in enhancing local green innovation levels.

## 6. Conclusions and Discussion

This paper measured the GIE of 30 provinces (autonomous regions and municipalities) in China by applying the Super-SBM model of non-expected output and the locational entropy formula, then empirically examined the spatial effects of MAGG on GIE using the SDM model with fixed effects. The findings show that (1) the overall GIE in China maintained a steady increase, and the level of MAGG slowly decreased during 2010–2019, with characteristics of obvious regional non-equilibrium and spatial correlations; (2) MAGG has a significant effect on the improvement of GIE nationally; (3) under the constraints of regional heterogeneity, the impacts of MAGG on GIE show significant differences between eastern, central, and western China; (4) in terms of industry heterogeneity, high-tech MAGG can significantly enhance local GIE, while the indirect effect of non-high-tech MAGG is significantly negative.

The policy implications based on this systematic theoretical and empirical analysis are as follows:

First, the level of MAGG should be enhanced reasonably. Local governments should moderately promote the development of manufacturing clusters in order to effectively exploit the knowledge and technology spillover effect, resource sharing effect, specialized division, and economy of scale effect, thereby enhancing the GIE. Furthermore, with the upgrading of agglomeration, importance should be attached to the staggered development of manufacturing industries, aiming to avoid the vicious competition of homogeneous industries for resources. Moreover, efforts should be made to achieve virtuous interaction between MAGG and GIE, ultimately realizing the high-quality development of the economy.

Second, the interconnection of MAGG between regions should be strengthened effectively. The results of the regional heterogeneity test revealed that the impact of MAGG on GIE was highly uneven across eastern, central, and western China due to differences in regional conditions, such as geographical location, resource endowment, and economic development. In this regard, the government should take local conditions into account and encourage the manufacturing industry in the eastern region to continue green innovation and concentrate on innovation input–output ratio, thereby maximizing the use of resources. As for the manufacturing industries in the central and western regions, the government should introduce more policies to stimulate enterprises to engage in green innovation, focus on the treatment of polluting enterprises, and regard talent introduction as a long-term strategy, thereby providing a constant impetus for the manufacturing industry. In order to realize inter-regional connectivity, manufacturing industries in various regions should share knowledge and technology, exchange experiences, and cooperate on projects, thereby promoting the coordinated development of green innovation technologies in China as a whole. Moreover, the local governments should try to break down the entry barriers and reduce intervention, so that the spillover of industrial agglomeration can take effect.

Third, the agglomeration layout of manufacturing sectors should be arranged scientifically. According to the results of the industry heterogeneity test, the impacts of MAGG on GIE were different between high-tech sectors and non-high-tech sectors. The government should take local industrial conditions into account in order to formulate different industrial policies. Since high-tech MAGG can create plenty of employment opportunities and attract talent, for regions with a low level of high-tech MAGG, the government should actively introduce more high-tech enterprises, extend preferential policies on taxation and loans in order to create high-tech MAGG with local characteristics, improve the external environment such as transportation and environmental protection in order to encourage the division and cooperation among enterprises in an agglomeration area, and transmit advanced environmental protection and energy-saving technologies in order to promote GIE. For regions with high levels of high-tech MAGG, the government should normalize competition and cooperation. If an enterprise is categorized as belonging to the high-tech sector but with low innovation efficiency and high pollution, it should be encouraged to transform and upgrade in a green manner, thus further amplifying the externalities of high-tech MAGG.

Fourth, the environment for green innovation should be optimized vigorously. Owing to the empirical results of the control variables, there are also several inspirations for the business environment. The implementation of strict environmental regulation policies by the government may sometimes place a burden on enterprises, thus making green innovation less effective. Therefore, the government should also explore the function of the market in solving environmental problems while managing environmental pollution by means of practical policies. Foreign investment should be introduced in a scientific and rational manner, so as to encourage the flow of foreign investment into green and innovative industries. Furthermore, the government should continue to improve the construction of basic transport facilities between regions. Regarding regions with relatively weak transport facilities, the cultivation of talent should be deemed a long-term strategy. By increasing investment in the construction of regional and inter-regional transport networks, the cost of inter-regional production factor mobility can be reduced, thus contributing to the joint development of a green economy.

Since many developing countries are facing the same pressing questions about balancing the environment and the economy, it is believed that these policy implications will also be conductive for them to realize green development. 

However, there are still some possible limitations: (1) this paper analyzes the impact mechanism of MAGG on GIE only by theoretical deduction, but it does not explore theoretical models, which can be carried out in our future research; (2) this paper uses 300 provincial-level panel data on China for empirical analysis, and does not consider the samples from other countries, so we can take cross-country panel data into account for international comparison in future research. 

## Figures and Tables

**Figure 1 ijerph-20-04238-f001:**
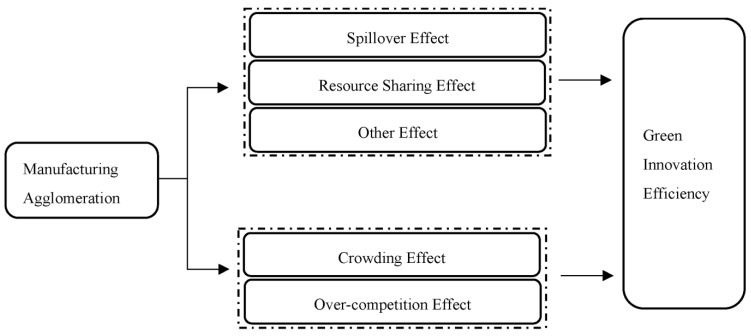
Effect Mechanism of MAGG on GIE.

**Figure 2 ijerph-20-04238-f002:**
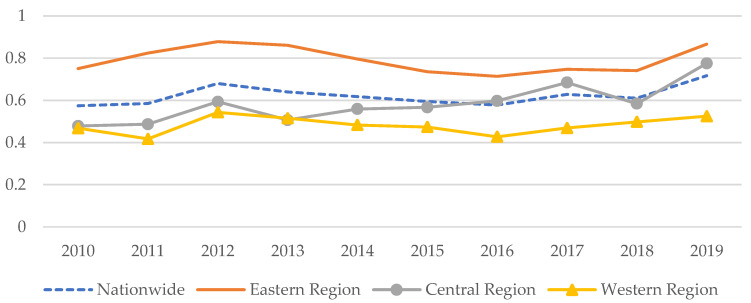
The trends of GIE levels during 2010–2019.

**Figure 3 ijerph-20-04238-f003:**
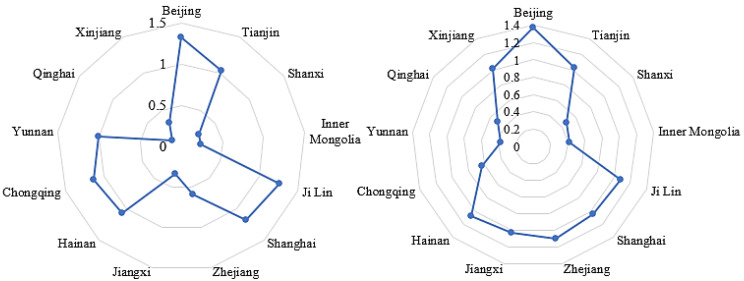
The GIE trends of selected provinces (autonomous regions and municipalities) in 2010 and 2019.

**Figure 4 ijerph-20-04238-f004:**
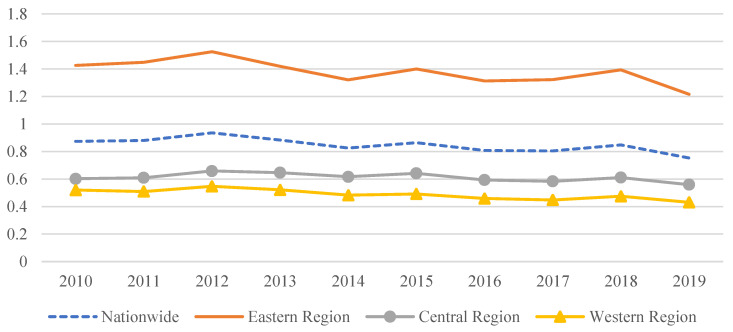
The trends of MAGG during the years 2010–2019.

**Figure 5 ijerph-20-04238-f005:**
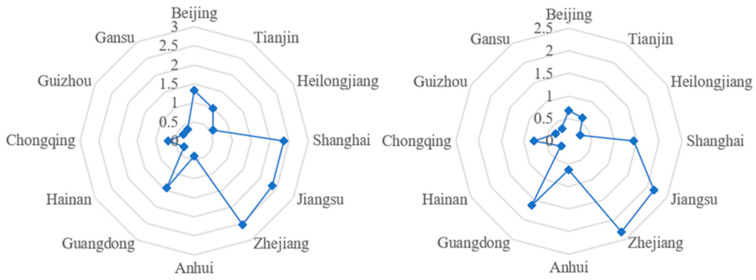
The MAGG trends of selected provinces (autonomous regions and municipalities) in 2010 and 2019.

**Figure 6 ijerph-20-04238-f006:**
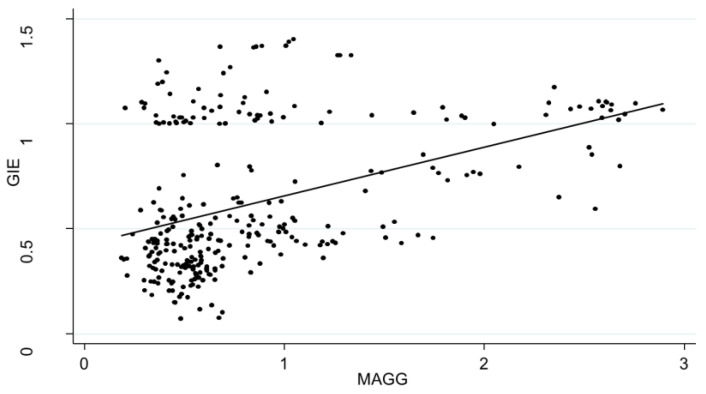
Relationship between MAGG and GIE in China during the years 2010–2019.

**Figure 7 ijerph-20-04238-f007:**
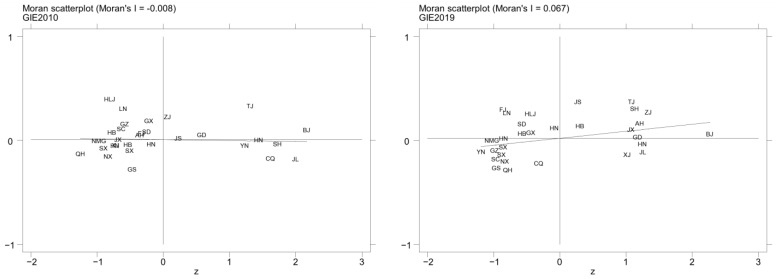
Local Moran’s *I* scatterplot of GIE of China in 2010 and 2019.

**Figure 8 ijerph-20-04238-f008:**
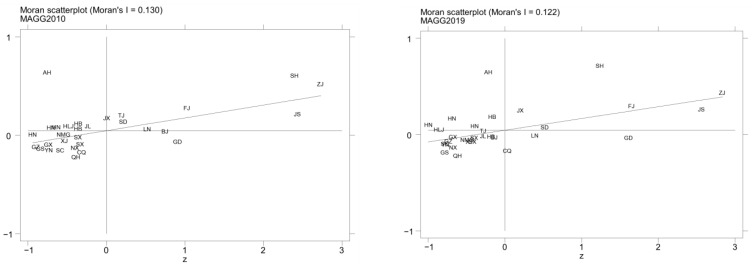
Local Moran’ *I* scatterplot of MAGG levels in China in 2010 and 2019.

**Figure 9 ijerph-20-04238-f009:**
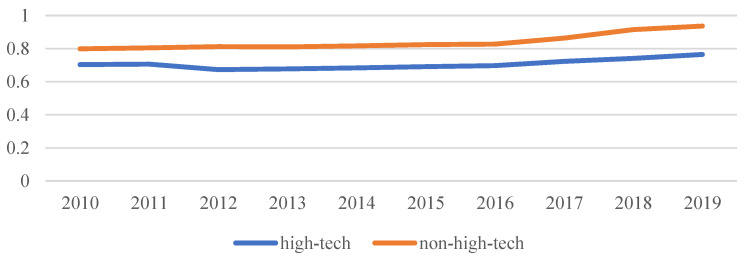
The trends of high-tech and non-high-tech MAGG levels during the years 2010–2019.

**Figure 10 ijerph-20-04238-f010:**
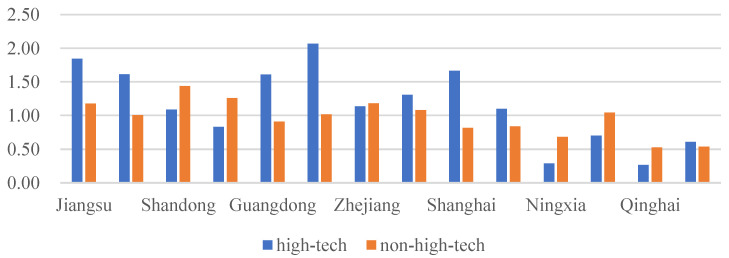
The MAGG level of representative provinces (autonomous regions and municipalities) in 2010 and 2019.

**Table 1 ijerph-20-04238-t001:** Input–output table for GIE.

Indicator Type	Indicator Category	Measurements
Input Indicators	Capital input	R&D expenditure by region
		New product development expenditure
	Labor input	Full-time equivalent of R&D personnel by region
	Energy input	Energy consumption by region
Output Indicators	Desired output	Number of green patents granted by region
		New product sales revenue
	Non-desired output	CO_2_ emissions by region

**Table 2 ijerph-20-04238-t002:** Description of variables.

Variable Type	Variable Meaning	Variable	Measurement Method	Unit	Data Source
Explained variable	Green Innovation Efficiency	GIE	Measured by Super-SEM model based on non-desired output	-	Science and Technology Statistical Yearbook, Industrial Statistics Yearbook, State Intellectual Property Office Patent Search System
Explanatory variable	Manufacturing Agglomeration	MAGG	Calculated by zone entropy formula	-	Labor Statistics Yearbook
Control variables	Environmental Regulation	ENV	Pollution control completion amount	ten thousand Yuan	China Statistical Yearbook
	Industrial Structure	IS	The proportion of the output value of tertiary industry to the GDP of the region	%	China Statistical Yearbook
	Level of External Opening	FD	The amount of actual foreign capital used as a proportion of the GDP of the region	%	China Statistical Yearbook
	Economic Development Level	GDP	GDP value of each region	Yuan	China Statistical Yearbook and regional statistical yearbooks
	Level of Traffic Infrastructure	TRA	Road kilometers per square kilometer	km/(km)²	China Statistical Yearbook

**Table 3 ijerph-20-04238-t003:** Descriptive statistics values of variables.

Variables	Variable Meaning	Sample Size	Mean	Standard Deviation	Min	Max
GIE	Green Innovation Efficiency	300	0.622	0.336	0.074	1.405
MAGG	Manufacturing Agglomeration	300	0.848	0.618	0.183	2.891
lnENV	Environmental Regulation	300	11.89	1.006	8.379	14.16
IS	Industrial Structure	300	0.457	0.0976	0.286	0.835
FD	Level of External Opening	300	0.0195	0.0155	4.24 × 10^−6^	0.0796
lnGDP	Economic Development Level	300	18.96	0.865	16.42	20.80
TRA	Level of Traffic Infrastructure	300	0.924	0.497	0.0862	2.125

**Table 4 ijerph-20-04238-t004:** Stepwise regression results.

	(1)	(2)	(3)	(4)	(5)	(6)
MAGG	0.232 ***	0.266 ***	0.202 ***	0.18 ***	0.153 ***	0.13 ***
	(0.029)	(0.03)	(0.029)	(0.029)	(0.03)	(0.03)
lnENV		−0.062 ***	−0.033 *	−0.037 **	−0.07 ***	−0.068 ***
		(0.018)	(0.017)	(0.017)	(0.021)	(0.021)
IS			10.237 ***	10.169 ***	10.064 ***	0.869 ***
			(0.175)	(0.17)	(0.173)	(0.175)
FD				40.726 ***	40.279 ***	30.076 ***
				(10.032)	(10.036)	(10.054)
lnGDP					0.07 ***	0.027
					(0.027)	(0.028)
TRA						0.172 ***
						(0.043)
_cons	0.426 ***	10.129 ***	0.277	0.281	−0.573	0.183
	(0.03)	(0.212)	(0.23)	(0.223)	(0.395)	(0.429)
Observations	300	300	300	300	300	300
R-squared	0.181	0.211	0.326	0.37	0.385	0.416

Note: Standard errors are in parentheses. *, **, *** indicate significant at the 10%, 5%, and 1% levels, respectively.

**Table 5 ijerph-20-04238-t005:** OLS and 2SLS estimation results.

	OLS	2SLS	
		GIS	POP
MAGG	0.13 ***	1.144 *	0.198 ***
	(0.03)	(0.585)	(0.05)
Control variables	YES	YES	YES
Observations	300	300	300
R-squared	0.416	0.35	0.406
Phase I F-statistic		26.61	28.78

Note: Standard errors are in parentheses. *, *** indicate significant at the 10% and 1% levels, respectively.

**Table 6 ijerph-20-04238-t006:** Global Moran’s *I* Index of GIE and MAGG.

	GIE				MAGG		
Year	Moran’s *I*	Z-Value	*p*-Value	Year	Moran’s *I*	Z-Value	*p*-Value
2010	−0.008	0.723	0.235	2010	0.130	4.734	0.000
2011	0.030	1.781	0.037	2011	0.140	5.034	0.000
2012	0.035	1.900	0.029	2012	0.142	5.153	0.000
2013	0.052	2.369	0.009	2013	0.112	4.258	0.000
2014	0.066	2.755	0.003	2014	0.111	4.232	0.000
2015	0.050	2.318	0.010	2015	0.113	4.305	0.000
2016	0.049	2.301	0.011	2016	0.113	4.326	0.000
2017	0.027	1.670	0.047	2017	0.117	4.390	0.000
2018	0.033	1.855	0.032	2018	0.120	4.453	0.000
2019	0.067	2.773	0.003	2019	0.122	4.500	0.000

**Table 7 ijerph-20-04238-t007:** Regression results of the impact of MAGG on GIE in China.

	(1)	(2)	(3)	(4)	(5)	(6)
	RE	FE	OLS	SEM	SAR	SDM
MAGG	0.172 ***	0.206 **	0.13 ***	0.109 ***	0.114 ***	0.093 ***
	(0.053)	(0.091)	(0.03)	(0.027)	(0.031)	(0.031)
lnENV	0.018	0.04 **	−0.068 ***	−0.086 ***	−0.073 ***	−0.071 ***
	(0.019)	(0.019)	(0.021)	(0.019)	(0.02)	(0.019)
IS	0.608 ***	0.386	0.869 ***	1.19 ***	1.174 ***	0.756 ***
	(0.229)	(0.323)	(0.175)	(0.192)	(0.202)	(0.218)
FD	1.425	0.222	3.076 ***	3.511 ***	3.186 ***	4.148 ***
	(1.13)	(1.238)	(1.054)	(1.077)	(1.09)	(1.222)
lnGDP	−0.055	0.009	0.027	0.058 **	0.061 **	−0.068 *
	(0.044)	(0.076)	(0.028)	(0.029)	(0.029)	(0.036)
TRA	0.194 **	−0.008	0.172 ***	0.164 ***	0.163 ***	0.248 ***
	(0.087)	(0.211)	(0.043)	(0.04)	(0.041)	(0.046)
Constant term	0.824	−0.359	0.183			
	(0.71)	(1.257)	(0.429)			
ρ				−1.175 ***	−0.471 **	−0.853 ***
				(0.26)	(0.23)	(0.253)
Variance:sigma2_e				0.055 ***	0.06 ***	0.05 ***
				(0.005)	(0.005)	(0.004)
Wx:MAGG						0.428 **
						(0.206)
Wx:lnENV						−0.533 ***
						(0.117)
Wx:IS						0.969
						(1.062)
Wx:FD						42.203 ***
						(8.008)
Wx:lnGDP						−0.577 **
						(0.245)
Wx:TRA						1.081 ***
						(0.326)
Hausmann Testing						21.00 *
						[0.073]
LR_spatial_lag						27.34 ***
						[0.000]
LR_spatial_error						29.44 ***
						[0.000]
Observations	300	300	300	300	300	300
R-squared	0.014	0.04	0.416	0.409	0.383	0.41

Note: ***, **, * denote significance at the 1%, 5%, and 10% levels, respectively, with standard deviation in small brackets and *p*-value in middle brackets.

**Table 8 ijerph-20-04238-t008:** Robustness test of the effect of MAGG on GIE.

	Economic Distance Matrix	Replacing Explanatory Variable	Adjusting Time Span
ρ	−0.208 *	−0.829 ***	−0.563 *
	(0.111)	(0.252)	(0.293)
MAGG	0.206 **	0.778 *	0.133 ***
	(0.102)	(0.471)	(0.037)
Wx:MAGG	0.284 **	0.46 ***	0.378 **
	(0.121)	(0.159)	(0.188)
R-squared	0.437	0.41	0.369

Note: Standard deviations in parentheses; *, **, *** indicate significant at the 10%, 5%, and 1% levels, respectively.

**Table 9 ijerph-20-04238-t009:** The results of spatial effect decomposition based on the SDM model.

GIE	Direct Effect		Indirect Effect		Total Effect	
	Coefficients	*p*-Values	Coefficients	*p*-Values	Coefficients	*p*-Values
MAGG	0.086 **	(0.01)	0.203	(0.12)	0.289 **	(0.01)
lnENV	−0.055 ***	(0.01)	−0.274 ***	(0.00)	−0.329 ***	(0.00)
IS	0.742 ***	(0.00)	0.228	(0.70)	0.970 *	(0.09)
FD	2.915 ***	(0.01)	22.762 ***	(0.00)	25.677 ***	(0.00)
lnGDP	−0.055 ***	(0.09)	−0.317 **	(0.04)	−0.372 **	(0.02)
TRA	0.222 ***	(0.00)	0.507 **	(0.01)	0.729 ***	(0.00)

Note: *p*-values are in parentheses; *, **, *** indicate significance at the 10%, 5%, and 1% levels, respectively.

**Table 10 ijerph-20-04238-t010:** Decomposition results of spatial effects in eastern, central, and western regions.

	Eastern Region		Central Region		Western Region	
	Direct Effect	Indirect Effect	Direct Effect	Indirect Effect	Direct Effect	Indirect Effect
MAGG	0.388 ***	−0.515 ***	−0.147	2.595 ***	0.035	−1.013
	(0.00)	(0.00)	(0.51)	(0.00)	(0.91)	(0.20)
lnENV	0.016	0.021	−0.085	−0.552 ***	0.002	0.102
	(0.63)	(0.80)	(0.12)	(0.00)	(0.96)	(0.42)
IS	0.812	−1.221	−1.265	2.970	−1.494 **	0.083
	(0.13)	(0.22)	(0.17)	(0.18)	(0.04)	(0.97)
FD	1.482	−2.752	−2.903	21.719 **	−10.725 **	63.922 ***
	(0.15)	(0.44)	(0.50)	(0.04)	(0.04)	(0.00)
lnGDP	−0.363 ***	−0.471	−0.646 ***	0.995 **	−0.014	−0.225
	(0.00)	(0.15)	(0.00)	(0.04)	(0.83)	(0.21)
TRA	0.362 **	0.679 *	1.086 ***	−1.573 ***	0.182 *	0.354
	(0.01)	(0.07)	(0.00)	(0.00)	(0.08)	(0.30)

Note: *p*-values are in parentheses; *, **, *** indicate significance at the 10%, 5%, and 1% levels, respectively.

**Table 11 ijerph-20-04238-t011:** Decomposition results of the spatial effects of high-tech and non-high-tech manufacturing sectors.

	High-Tech Manufacturing Sectors			Non-High-Tech Manufacturing Sectors		
	Direct Effect	Indirect Effect	Total Effect	Direct Effect	Indirect Effect	Total Effect
MAGG	0.275 ***	0.248	0.523 ***	0.112	−0.769 **	−0.657 **
	(0.00)	(0.20)	(0.01)	(0.12)	(0.01)	(0.03)
lnENV	−0.042 **	−0.183 ***	−0.226 ***	−0.058 ***	−0.203 ***	−0.261 ***
	(0.03)	(0.01)	(0.00)	(0.00)	(0.01)	(0.00)
IS	1.136 ***	0.803	1.939 ***	1.580 ***	−1.225	0.355
	(0.00)	(0.25)	(0.00)	(0.00)	(0.12)	(0.62)
FD	1.605	18.545 ***	20.150 ***	1.766	17.777 ***	19.543 ***
	(0.12)	(0.00)	(0.00)	(0.11)	(0.00)	(0.00)
lnGDP	−0.076 **	−0.217 *	−0.292 **	−0.008	0.060	0.052
	(0.01)	(0.09)	(0.03)	(0.81)	(0.67)	(0.73)
TRA	0.106 **	0.385 *	0.491 **	0.123 **	0.707 ***	0.831 ***
	(0.02)	(0.07)	(0.02)	(0.02)	(0.00)	(0.00)

Note: *p*-values are in parentheses; *, **, *** indicate significance at the 10%, 5%, and 1% levels, respectively.

## Data Availability

Some or all data and models that support the findings of this study are available from the corresponding author upon reasonable request.

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
