# Peer review of "Impact of Manufacturing Agglomeration on the Green Innovation Efficiency—Spatial Effect Based on China’s Provincial Panel Data"

_ijerph, 2023, doi:10.3390/ijerph20054238_

Round 1
Reviewer 1 Report
Using China’s province-level data, this paper investigates the effect of manufacturing agglomeration on green innovation efficiency. The result reveals that industrial agglomeration has a great enhancement effect on provincial green innovation efficiency. The topic discussed in this paper is interesting and is novel to the body of existing literature. But there remain some shortcomings. The comments are as follows.
1. In the introduction part, the author shows that “Presently, China's manufacturing industry is still in the mode of high input, high 44consumption, and high pollution, accompanied with low energy efficiency and numerous 45prominent resource and environmental problems.” You’d better give some evidence.
2. The author should add more information about this paper’s linkage to other research. Why your paper is important? Talk more about the contribution of this paper.
3. There are many “Error! Reference source not found.”, what’s wrong?
4. Why you use MAGG to represent agglomeration level in this paper. You can give the reason. And you should use other index to prove the robustness of your result.
5. In Figure 4, the range of horizontal axis is 1-10. Refer to the contents, it may be wrong. It should be year.
6. In this paper, agglomeration intensity is measured at province level, so all the regression standard errors should be clustered at province level.
7. The endogeneity of agglomeration intensity is an important issue in this paper. It will bias the estimation results. You’d better find a good instrument variable to solve this problem.
8. In Table 3, all the measuring units of the variables should be added.
9. In the conclusion part, the policy recommendations you introduced should be closely related to the results you got.
10. The English writing of this paper needs to be further proofed.
Reviewer 2 Report
The paper contains good information and deals with an interesting issue about the manufacturing agglomeration and green innovation efficiency. However, the paper has some flaws, which need serious considerations:
Abstract
ï‚· The abstract is well-written and attracts the attention of readers. Nevertheless, may be few structural modifications can make it more attractive. The authors may begin the abstract with the purpose of the study. Next, the significance or background of the study can be provided. The methodology of the study should be presented. Then, the findings and, finally, 2-3 implications can boost the attractiveness of the study.
Introduction
ï‚· In introduction, authors need to highlight more about the unique features of the Chinese case that can contribute significantly to our existing knowledge regarding the impact of manufacturing agglomeration on green innovation. Why do people need to know about this?
ï‚· The impact of manufacturing agglomeration on green innovation efficiency have already been noticed elsewhere in the standard literature, why are re-examining these factors for China is important and what kind of new insights can be revealed about green innovation efficiency by doing so? These questions need to be addressed carefully.
ï‚· The introduction needs more highlighting about the innovative and novelty of the study and the research problem [gap(s)].
ï‚· You may add a description of the structure of the paper
literature review
ï‚· The author’s may bring the relevant theories to support the hypothesis that manufacturing agglomeration has an impact on green innovation efficiency.
ï‚· In the literature review and study in general, authors need to employ current studies on their topic since the reported prior studies are not up to date. Most of the studies are too old.
Methodology and Results
ï‚· The authors used the common measurements for the study variables (Manufacturing Agglomeration Level and Green Innovation Efficiency)
ï‚· The authors followed the common methods for analysis.
However:
ï‚· The study used data from 2010 to 2019 but there was no justification for the period chosen.
ï‚· I missed the discussion section. The authors did not do discussion of the results with connection to the theories and previous literature, hence the application to the study was not robust enough. In this section, the authors should further expand their work, highlighting the similarities and differences with the existing studies and provide some more critical and insightful arguments on the topic to challenge the current situation and call for further developments.
ï‚· The empirical results and discussion are not explicitly discussed in relation to the Chinese case.
Conclusions and Recommendations
ï‚· Authors should mention the limitations and future directions of their study.
Further comments:
- There were instances where the authors referred to “past literature or studies” but no particular studies were referenced. for example, page 2 line 62.
Round 2
Reviewer 1 Report
The authors have made a commendable effort to address most of reviewers’ concerns within the available space. I am satisfied with the revised version of the paper. I recommend published in this journal.